# Knowledge about traumatic World War II experiences among ancestors and subjective well-being of young adults: A person-centred perspective

**Marcin Rzeszutek**[1]*, **Maja Lis-Turlejska**[2], **Małgorzata Pięta**[1], **Monika Karlsen**[1], **Holly Backus**[1], **Wiktoria Florek**[1], **Katarzyna Lisowska**[1], **Daniel Pankowski**[3], **Szymon Szumiał**[2]

**1** Faculty of Psychology, University of Warsaw, Warsaw, Poland, **2** Faculty of Psychology, SWPS University of Social Sciences and Humanities, Warsaw, Poland, **3** Faculty of Psychology, University of Economics and Human Sciences in Warsaw, Warsaw, Poland

* marcin.rzeszutek@psych.uw.edu.pl

**Data Availability Statement:** Data are included in the Supporting Information.

## Abstract

### Objectives

The aim of our study is to examine the association between knowledge about the World War II (WWII)-related traumatic experiences of their ancestors and subjective well-being (SWB) of young adults, i.e., descendants of Polish survivors of WWII. Specifically, we focus on the life satisfaction and the mental, physical, and psychosocial well-being of our participants in relation to their knowledge about WWII trauma in their family histories.

### Method

The sample comprised 500 Polish young adults recruited from a nonclinical general population. Participants first filled out a questionnaire assessing their knowledge about traumatic events that their ancestors could have experienced during WWII (see grandparents/mothers, great-grandparents/mothers). After that, subjects were given inventories to assess their SWB, i.e., the Satisfaction with Life Scale (SWLS) and the General Health Questionnaire (GHQ-28).

### Results

Latent profile analysis was applied to extract profiles of participants differing with regard to the scope of knowledge about WWII-related traumatic experiences among ancestors. Specifically, six profiles were observed, and a general lack of knowledge about this kind of trauma in the family was characteristic of the sample. We also found differences in SWB across profiles of participants, with worse SWB in the profiles with the highest lack of knowledge about WWII-related traumatic experiences in the family.

### Conclusion

Our study adds to the literature on intergenerational trauma by applying a person-centred perspective, a methodological approach almost invisible in research on that topic. In

**Funding:** The study was co-financed from the internal funds of the Faculty of Psychology, University of Warsaw (grant recipient Marcin Rzeszutek), the Faculty of Psychology, SWPS University of Social Sciences and Humanities (grant recipient Maja Lis-Turlejska) and the Faculty of Psychology, University of Economics and Human Sciences in Warsaw (grant recipient Daniel Pankowski). The funders had no role in study design, data collection and analysis, decision to publish, or preparation of the manuscript.

**Competing interests:** The authors have declared that no competing interests exist.

addition, our findings can serve as a stimulus for more comprehensive debate on WWII trauma in Polish society.

## Introduction

More than half a century ago, Rakoff et al. [1] observed and described the phenomenon of *intergenerational trauma* manifesting in psychopathological symptoms among descendants of Holocaust survivors—descendants who did not experience this kind of massive traumatization directly, but were *secondarily traumatized* by parental traumatic history. More specifically, it was found that children and grandchildren of Holocaust survivors, i.e., of the second and third generation, reported considerable levels of depression and anxiety, constant guilt and shame, difficulties in independent social functioning, and various physical health problems like cancer, heart disease, or chronic pain [see for review 2]. Since then, research on the long-lasting impact of trauma passed from one generation to another has grown enormously, including primarily further research on adult offspring of Holocaust survivors [3–7], but also families of war veterans (e.g., World War II, e.g., [8–10]), refugees [11], and even survivors of childhood maltreatment and abuse [12]. Nevertheless, studies on intergenerational trauma have also aroused controversy, as several questions remained unanswered. Specifically, it is still not known whether trauma transmission imposes negative clinical consequences on subsequent generations or not [e.g., compare [7, 13, 14] and which generation (the second or third) is affected most by the traumatic experiences of ancestors [e.g., compare [15, 16]]. Moreover, no convincing mechanism that underlies the transmission of trauma through subsequent generations has been provided, and existing theories about this phenomena range from genetic and neurobiological factors [7] to psychosocial explanations pointing to adverse parenting styles and drawbacks in family communication [17]. Additionally, research on the intergenerational trauma was conducted predominantly in the *variable-centred approach* neglecting how particular subgroups of participants may cluster across the studied predictors and outcomes, which may be the possible reason for many conflicting results in this field [see for review 2, 11]. Finally, studies on intergenerational trauma are unequally distributed in different parts of Europe, with only a few studies available from Central and Eastern Europe [18]. In our project we concentrate on the link between knowledge about World War II (WWII)-related traumatic experiences of their ancestors and subjective well-being among young adults in Poland, i.e., descendants of Polish survivors of WWII.

The problem of the psychological consequences of WWII in Poland is greatly omitted both in scientific research as well as in public debate [19]. Aforementioned fact precludes understanding the significant differences between the prevalence of WWII-related psychopathology, particularly posttraumatic stress disorder (PTSD), among survivors of WWII in Poland compared to those in other European countries. Specifically, studies conducted among civilian survivors of WWII in Western Europe point to the prevalence of a PTSD level ranging from 1.9% in Austria [20] to 4.6% in the Netherlands [21] to 10.9% in Germany [22]. In contrast, similar research projects carried out in Poland show significantly higher PTSD intensity among Polish survivors of WWII, varying from 29.4% [23] to even 38.3% [19]. In order to explain this huge discrepancy, it is vital to underline first historical factors, i.e., the extraordinary intensity and prevalence of various WWII-related traumatic events in Poland. For example, during WWII Poland lost about 17% of its pre-war population, which was the highest percentage among all countries taking part in this war [24]. Many groups of Polish people faced multiple traumas, to mention only war veterans of various armed forces and resistance movements, persons

deported deep into the Soviet Union in 1942–1944, the inhabitants of Warsaw as well as of many other towns who lost their loved ones, witnessed the executions, bombings, and fires, and witnesses of the Holocaust [25]. In addition, the sociopolitical situation in Poland after WWII, i.e., during the Communist regime (1946–1989), when many Poles were experiencing repression and insecurity, created significant obstacles that precluded revealing traumatic experiences to other people and obtaining support and social acknowledgment of their WWII trauma [19, 26]. Importantly, this problem was also present in families of Polish WWII survivors, where special patterns of communication can be observed characterized by an atmosphere of secrecy and taboo around the WWII-related traumatic experiences [19, 23, 27].

According to some theories, one of the main *vehicles* of intergenerational trauma transmission is the family, which may be associated with heritable epigenetic changes through subsequent generations [28], but also with specific patterns of attachment and communication [29]. More specifically, several studies conducted among children of Holocaust survivors reveal the link between perceived parental burden and parental overprotectiveness with the already-mentioned psychopathological and physical health problems of these children [17, 30, 31]. In addition, the conspiracy of silence, i.e., the ban on talking about any details of the Holocaust [6], which was very characteristic of the atmosphere of the family life of Polish survivors of WWII, contributed greatly to the transmission of trauma. Thus, the second and third generations of Holocaust survivors could not build a coherent narration of their family history of trauma [31]. According to the developmental perspective, sharing family history with children is a crucial element of the creation of their self-identity [32]. Furthermore, knowledge about family history is systematically mentioned as the major predictor of psychological well-being of young adults [33]. However, this topic has been poorly examined with regard to the knowledge of traumatic family stories, especially those experienced by past generations [34].

## Current study

Taking these research gaps into consideration, the aim of our study is to examine the association between knowledge about WWII-related traumatic experiences of their ancestors and subjective well-being (SWB) among young adults in Poland, i.e., descendants of Polish survivors of WWII. Specifically, we focus on the satisfaction with life and the mental, physical, and psychosocial well-being of our participants in relation to their knowledge about WWII trauma in their family histories. The novelty of our study is the application of the *person-centred perspective*, as until now studies on intergenerational trauma have focused only on the variable-centred approach, which neglects the problem of *heterogeneity* of participants with regard to the studied predictors and outcomes [see e.g., for review, 2, 11]. Thus, although our study is as such mainly explorative, we formulate two hypotheses. First, we expect that the studied sample will be heterogeneous in terms of knowledge about WWII-related traumatic experiences among ancestors, i.e., different profiles can be observed among the participants with regard to this aspect. Second, we assume that belonging to a particular profile will be differently related to subjective well-being among participants (i.e., satisfaction with life and mental, physical, and psychosocial well-being), after controlling for sociodemographic correlates. Particularly, we presume that *lack of knowledge* about WWII-related traumatic experiences among ancestors will be related to worse SWB among our participants.

## Method

### Participants and procedure

The sample comprised of 500 Polish young adults (368 women and 132 men) recruited from a nonclinical general population, who were the third or the fourth generation of the Polish

survivors from the WWII. More specifically, out of 560 participants invited to the study, we excluded 60 because of a high level of missing data that precluded statistical analysis. Participants' ages varied from 18 to 35 ($M$ = 21.89, $SD$ = 3.59). In terms of education levels, 40 participants (8.0%) completed higher education, 165 participants (33.0%) had some higher education, 290 participants (58.0%) had secondary education, 3 participants (0.6%) had vocational education, and 3 participants (0.4%) had primary education. With regard to marital status, 267 participants (53.4%) were single, 27 participants (5.4%) were married, 194 participants (38.8%) were involved in an informal relationship, 2 participants (0.4%) were separated, and10 participants (0.2%) were divorced.

The study was anonymous, and participation was voluntary. Informed consent was obtained from all participants before they were included in the research, and the participants did not receive remuneration for taking part in the project. All procedures performed in studies involving human participants were in accordance with the ethical standards of the institutional and/or national research committee and with the 1964 Helsinki declaration and its later amendments or comparable ethical standards. Our research project was accepted by the ethics committee of the Faculty of Psychology, University of Warsaw.

## Measures

**Knowledge about traumatic WWII experiences in the family.**　In the first part of the study, participants filled out the questionnaire, which checked their knowledge of traumatic events that their ancestors may have experienced during WWII (grandparents/mothers, great grandparents/mothers; [19]). Twenty-nine questions concerned the WWII-related traumatic experiences of the ancestors of the participants on the part of the mother, as well as the father, of each. Therefore, the respondent had to answer questions about the same traumatic events in relation to four people from the generation who survived WWII. Specifically, respondents indicated whether according to their knowledge each event had happened or had not happened in the life of their grandparents, or whether they did not know. The number of events not known is the variable used in the current study. The questionnaire also asks about the grandparents'/mothers', great-grandparents'/mothers' approximate year of birth.

**Subjective well-being.**　Subjective well-being (SWB) was assessed first with the aid of the Satisfaction with Life Scale (SWLS) [35] in a Polish adaptation. The SWLS consists of five items, each with a 7-point scale, ranging from 1 (*strongly disagree*) to 7 (*strongly agree*). A higher total score means a higher level of satisfaction with life. The Cronbach's alpha in the current study was .85.

The second inventory was the General Health Questionnaire (GHQ-28) [36] in a Polish adaptation, which measures various aspects of mental, physical, and psychosocial well-being, including somatic symptoms, anxiety and insomnia, social dysfunction, and depressive symptoms. Participants are asked to assess changes in their mood, feelings, and behaviours in the period of the past four weeks on a 4-point Likert scale. The higher the score, the poorer the well-being of the participant. The Cronbach's alphas in the current study for the particular subscales were .78 for the A somatic symptoms subscale, .85 for the B anxiety/insomnia subscale, .85 for the C social dysfunction subscale, .91 for the D depression subscale, and .93 for the total score.

## Data analysis

The data analysis consisted of three steps. First demographic characteristics of the analysed sample, descriptive statistics, and intercorrelations between analysed variables were computed. The deviation from normality was assessed with the use of skewness and kurtosis measures.

The bootstrap method with the number of samples equal to 1.000 was used in the correlation analysis. In the next step, latent profile analysis was performed in order to extract potential subgroups of respondents and to verify the first hypothesis. The fit of models was assessed with the use of the Aikake information criterion (AIC) and Bayesian information criterion (BIC). The acquired profiles were centred in order to foster clear interpretation. The final stage was the analysis of mediation, including analysis of the relationship between knowledge about WWII-related traumatic events and subjective well-being, which verified the second hypothesis.

Descriptive statistics of the sample and analysed variables as well as the correlation analysis were performed with the use of IBM SPSS Statistics 25.0 software. The latent profile analysis was computed with the use of the tidyLPA package working in the R Statistics 3.6.2 environment [37]. Finally, the mediation was performed with the use of the Hayes [38] macro process version 3.4.

## Results

Table 1 presents the descriptive statistics for all analysed interval variables. The values of skewness and kurtosis that falling outside the interval from -1 to 1 indicated that the distribution of participants' age, number of events not remembered, and level of depression differed significantly from the normal distribution. As a consequence, the subsequent analysis was performed using the bootstrap method.

Table 1 also presents bootstrap confidence intervals for intercorrelations between analysed variables. Statistically significant correlations are marked in bold. Participants' ages correlated negatively with levels of anxiety/insomnia and depression. All indicators of lack of memory, i.e., number of events not known referring to maternal grandmother, maternal grandfather, paternal grandmother, and paternal grandfather correlated positively with each other. All

**Table 1. Descriptive statistics and pearson correlation coefficients between analysed variables acquired with the use of the bootstrap method.**

| | Variables | M | SD | S | K | 1. | 2. | 3. | 4. | 5. | 6. | 7. | 8. | 9. | 10. |
|---|---|---|---|---|---|---|---|---|---|---|---|---|---|---|---|
| | 1. Age | 21.89 | 3.59 | 2.93 | 12.11 | - | | | | | | | | | |
| Number of Events not Known | 2. Maternal Grandmother | 2.92 | 4.09 | 1.90 | 4.62 | -.09÷.08 | - | | | | | | | | |
| | 3. Maternal Grandfather | 2.93 | 4.64 | 2.00 | 4.40 | -.09÷.09 | **.41÷.64** | - | | | | | | | |
| | 4. Paternal Grandmother | 2.31 | 3.97 | 2.23 | 5.44 | -.12÷.06 | **.27÷.50** | **.26÷.51** | - | | | | | | |
| | 5. Paternal Grandfather | 2.78 | 4.82 | 2.08 | 4.12 | -.09÷.07 | **.30÷.51** | **.33÷.57** | **.42÷.66** | - | | | | | |
| | 6. Life Satisfaction | 21.28 | 5.97 | -.34 | -.22 | -.15÷.04 | -.13÷.10 | -.11÷.10 | -.04÷.17 | -.06÷.16 | - | | | | |
| Subjective-Well-Being | 7. Somatic Symptoms | 8.60 | 4.15 | .43 | -.39 | -.09÷.08 | -.06÷.13 | -.03÷.16 | -.05÷.16 | -.10÷.12 | **-.38÷-.22** | - | | | |
| | 8. Anxiety/ Insomnia | 8.61 | 4.72 | .45 | -.14 | **-.18÷-.01** | -.02÷.19 | -.02÷.19 | -.04÷.14 | -.09÷.12 | **-.38÷-.21** | **.55÷.68** | - | | |
| | 9. Social Dysfunction | 8.41 | 3.57 | .58 | .79 | -.10÷.06 | -.09÷.12 | -.06÷.15 | -.13÷.08 | -.09÷.14 | **-.47÷-.31** | **.40÷.55** | **.44÷.60** | - | |
| | 10. Depression | 4.24 | 4.82 | 1.52 | 1.85 | **-.18÷-.02** | -.07÷.16 | -.07÷.16 | -.09÷.10 | -.09÷.10 | **-.56÷-.42** | **.33÷.50** | **.43÷.59** | **.40÷.57** | - |
| | 11. GHQ-28 Total Score | 29.86 | 13.79 | .84 | .70 | -.16÷.01 | -.06÷.18 | -.04÷.19 | -.08÷.13 | -.10÷.13 | **-.54÷-.39** | **.75÷.82** | **.82÷.87** | **.70÷.80** | **.74÷.82** |

*M*–Mean Value; *SD*–Standard Deviation; *S*–Skewness; *K*–Kurtosis; * $p < .05$; ** $p < .01$.

scales of GHQ-28 also correlated positively with each other. Subjective well-being correlated negatively with the scores on all scales of GHQ-28.

Table 2 presents the frequency distribution for types of WWII-related traumatic events referring to which participants did not know if they had happened or not in the life of their maternal and paternal grandmothers and grandfathers.

The percentage of the respondents who did not know if the traumatic events had happened was considerably high. The lowest mean percentage was acquired referring to being forcibly relocated to Siberia and to being imprisoned in a Nazi concentration camp. The highest mean percentage was acquired referring to witnessed rape or other forms of sexual abuse.

In the process of the preparation of descriptive statistics, intercorrelations between analysed variables and frequency distributions for the events of which the respondents had no knowledge was the first step of performed statistical analysis. In the next step, latent profile analysis was executed in order to estimate distinct profiles and extract different subgroups of

**Table 2. Frequency distribution–types of WW-II-related traumatic events not known by respondents.**

| | Maternal | | Maternal | | Paternal | | Paternal | | Mean |
| | Grandmother | | Grandfather | | Grandmother | | Grandfather | | Percentage |
| Type of WW-II Related Traumatic Event | N | % | N | % | N | % | N | % | % |
|---|---|---|---|---|---|---|---|---|---|
| Loss of One's Mother | 252 | 50.4 | 328 | 65.6 | 319 | 63.8 | 350 | 70.0 | 62.5 |
| Loss of One's Father | 269 | 53.8 | 331 | 66.2 | 327 | 65.4 | 351 | 70.2 | 63.9 |
| Loss of One's Close Relative | 322 | 64.4 | 358 | 71.6 | 347 | 69.4 | 374 | 74.8 | 70.1 |
| Being in Combat | 227 | 45.4 | 275 | 55.0 | 297 | 59.4 | 328 | 65.6 | 56.4 |
| Being in Resistance | 239 | 47.8 | 292 | 58.4 | 308 | 61.6 | 338 | 67.6 | 58.9 |
| Being Wounded | 257 | 51.4 | 296 | 59.2 | 347 | 69.4 | 356 | 71.2 | 62.8 |
| Killed Someone | 238 | 47.6 | 314 | 62.8 | 298 | 59.6 | 348 | 69.6 | 59.9 |
| Being Tortured | 247 | 49.4 | 309 | 61.8 | 314 | 62.8 | 348 | 69.6 | 60.9 |
| **Being Imprisoned in a Nazi Concentration Camp** | **178** | **35.6** | **237** | **47.4** | **271** | **54.2** | **290** | **58.0** | **48.8** |
| Being Imprisoned in a Soviet Camp | 191 | 38.2 | 249 | 49.8 | 265 | 53.0 | 294 | 58.8 | 50.0 |
| Being in a Ghetto | 188 | 37.6 | 244 | 48.8 | 270 | 54.0 | 296 | 59.2 | 49.9 |
| Being in Warsaw During the Warsaw Uprising | 192 | 38.4 | 239 | 47.8 | 278 | 55.6 | 293 | 58.6 | 50.1 |
| Experienced Rape or Other Form of Sexual abuse | 310 | 62.0 | 312 | 62.4 | 353 | 70.6 | 347 | 69.4 | 66.1 |
| Surviving Bombing | 274 | 54.8 | 328 | 65.6 | 344 | 68.8 | 356 | 71.2 | 65.1 |
| Had to Remain in Hiding | 251 | 50.2 | 300 | 60.0 | 319 | 63.8 | 338 | 67.6 | 60.4 |
| Hiding Jews | 285 | 57.0 | 324 | 64.8 | 340 | 68.0 | 371 | 74.2 | 66.0 |
| **Being Forcedly Relocated to Siberia** | **185** | **37.0** | **248** | **49.6** | **262** | **52.4** | **284** | **56.8** | **49.0** |
| Being in Forced Labor | 211 | 42.2 | 266 | 53.2 | 267 | 53.4 | 298 | 59.6 | 52.1 |
| Health or Life Threatening Cold | 302 | 60.4 | 342 | 68.4 | 351 | 70.2 | 375 | 75.0 | 68.5 |
| Life Threatening Hunger | 293 | 58.6 | 343 | 68.6 | 351 | 70.2 | 366 | 73.2 | 67.7 |
| Fight in the resistance | 257 | 51.4 | 304 | 60.8 | 314 | 62.8 | 341 | 68.2 | 60.8 |
| Serious Illness | 305 | 61.0 | 340 | 68.0 | 366 | 73.2 | 375 | 75.0 | 69.3 |
| The Closest Person Subjected to Torture, Sexual Violence or Serious Injury | 338 | 67.6 | 380 | 76.0 | 386 | 77.2 | 396 | 79.2 | 75.0 |
| Witnessed Combat | 307 | 61.4 | 332 | 66.4 | 339 | 67.8 | 346 | 69.2 | 66.2 |
| Witnessed Somebody Being Shot | 345 | 69.0 | 362 | 72.4 | 378 | 75.6 | 379 | 75.8 | 73.2 |
| Witnessed Execution or Murder | 364 | 72.8 | 384 | 76.8 | 389 | 77.8 | 396 | 79.2 | 76.7 |
| **Witnessed Rape or other Form of Sexual Abuse** | **394** | **78.8** | **416** | **83.2** | **410** | **82.0** | **424** | **84.8** | **82.2** |
| Witnessed Somebody Being Heavily Beaten | 361 | 72.2 | 384 | 76.8 | 389 | 77.8 | 402 | 80.4 | 76.8 |
| Witnessed Assault or Persecution of Jews | 356 | 71.2 | 383 | 76.6 | 383 | 76.6 | 397 | 79.4 | 76.0 |

*N*–Number of Participants; %—Percent of the Sample.

respondents differing in regard to the number of events they did not know about in the lives of their maternal and paternal grandparents. We analysed four variables: the number of events not known regarding maternal grandmother, the number of events not known regarding maternal grandfather, the number of events not known regarding paternal grandmother, and the number of events not known regarding paternal grandfather.

According to the values of AIC and BIC, the model with best fit was the model with equal variances and covariances fixed to 0 and with 6 extracted classes with 6 distinctive profiles. The values of fit statistics were equal to AIC = 4,774.04 and BIC = 4,914.12. Fig 1 presents acquired profiles.

Class 5 was the class of respondents who unambiguously had the highest level of ignorance about WWII-related traumatic events in the lives of their grandparents. In Class 3, the level of lack of knowledge was unambiguously the lowest. Class 1 represents the group of respondents with an average level of knowledge. Class 2 is the group of respondents who knew much more about their paternal grandfathers. Class 4 represents lack of knowledge about paternal grandparents regarding WWII-related traumatic events. Class 6 represents lack of knowledge about paternal grandfathers. The extracted latent profiles confirm the first hypothesis.

The two extreme classes, i.e., class 3, representing the highest level of knowledge, and class 5, representing the highest level of lack of knowledge, were also analysed as explaining variables in the statistical model of mediation. The analysed model is depicted in Fig 2.

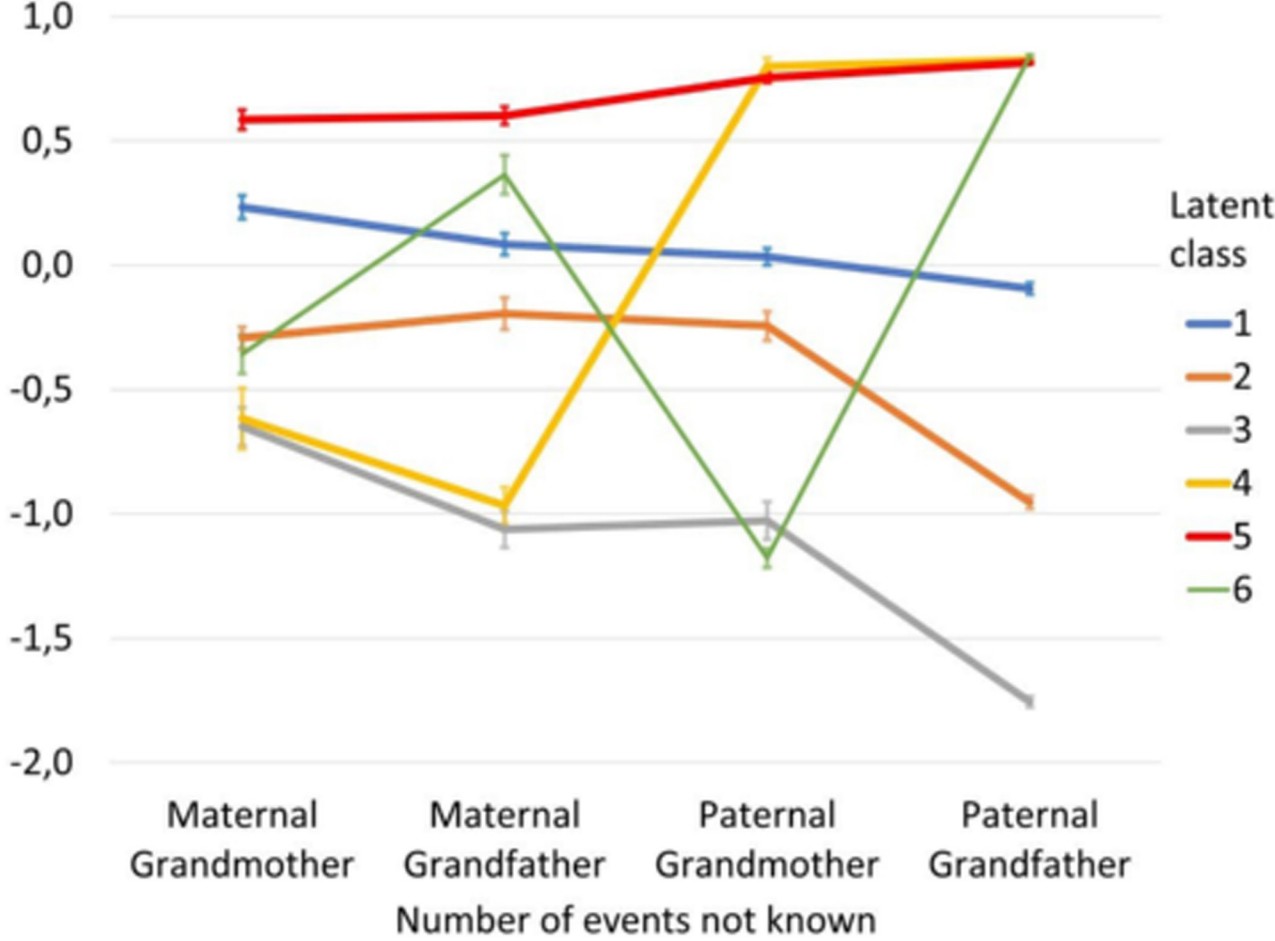

**Fig 1. Centered values of estimates of the analysed variables values in extracted classes with standard errors.**

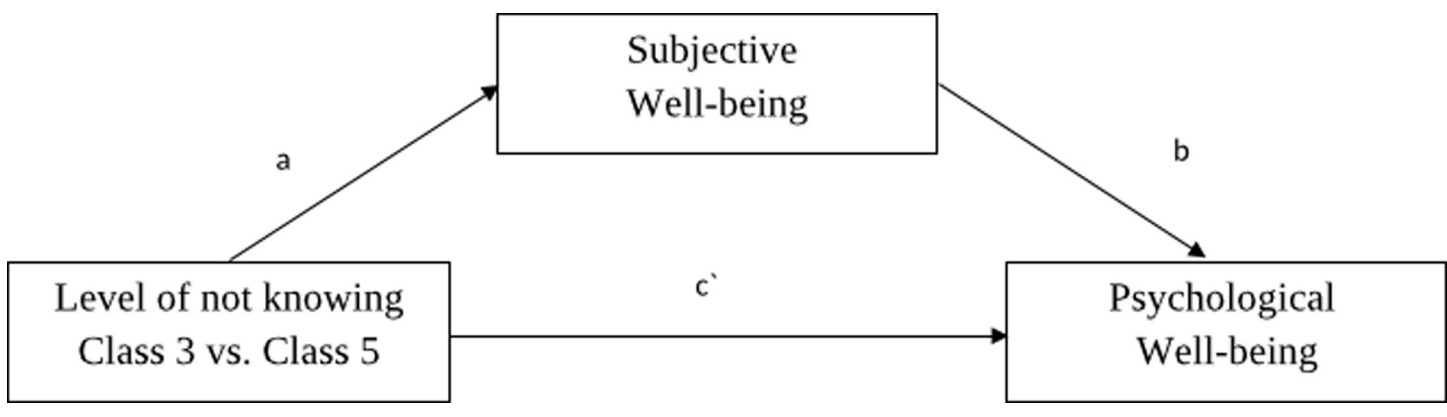

**Fig 2. Analysed statistical model of mediation.**

The level of ignorance about traumatic WWII events was analysed as the explaining variable, the level of life satisfaction as a mediator, and the GHQ-28 scales (somatic symptoms, anxiety/insomnia, social dysfunction, depression, and total score) being as the explained variable. Each scale was analysed in a separate statistical model, yielding five separate models. The acquired results are presented in Table 3.

In all five models, gender, age, and being in a relationship—formal or informal—were analysed as possible covariates. However, they were excluded if no statistical significance was found. Those significantly related to mediator or explained variable covariates were retained.

In all five models there was a statistically significant effect of mediation; the level of life satisfaction mediated the relationship between the level of ignorance and GHQ-28 scores in all five models. The participants from class 5 (with the highest level of ignorance) had lower levels of subjective well-being than participants from class 3 (with the lowest level of ignorance), which confirms the second hypothesis. The lower level of life satisfaction in consequence was associated with higher level of symptoms in GHQ-28. It refers to all scales analysed as explained variables; however, the effect was stronger regarding social dysfunction and depression. Considering covariates, it is worth mentioning that participants' gender was controlled for in the models regarding somatic symptoms. The level of somatic symptoms was significantly lower in the group of females compared to the group of male participants: $B = -0.35$, $t = -2.38$, $p < .05$. Participants' age was controlled for in the model regarding anxiety/insomnia. Age was negatively related to the level of symptoms: $B = -0.04$, $t = -2.22$, $p < .05$. Both

**Table 3. Relationships between the level of not knowing about the WW-II related traumatic events and GHQ-28 scores mediated by the life satisfaction level.**

| Mediator | Explained variable | Relationships | | | Indirect effect | df | $R^2$ |
|---|---|---|---|---|---|---|---|
| | | Path a | Path b | Path c' | | | |
| Life Satisfaction | Somatic Symptoms | -.43** | -.28*** | .02 | .03÷.24 | 3,239 | 0,33 |
| | Anxiety/Insomnia | -.43** | -.27*** | -.04 | .03÷.23 | 3,239 | 0,29 |
| | Social Dysfunction | -.43** | -.38*** | .10 | .05÷.29 | 2,239 | 0,40 |
| | Depression | -.43** | -.52*** | -.09 | .07÷.38 | 2,240 | 0,48 |
| | GHQ-28 Total Score | -.43** | -.45*** | -.04 | .06÷.34 | 4,237 | 0,49 |

df–Degrees of Freedom; $R^2$ –Variance Explained; Level of not Knowing About the WW-II Related Traumatic Events Was the Explained Variable in Each Model;* $p < .05$;

** $p < .01$.

Standardized estimates.

participants' gender and age were controlled for in the model regarding total GHQ-28 score. Again, the level of symptoms was significantly lower in the group of females than in the group of male participants: $B$ = -0.33, $t$ = -2.43, $p < .05$; and age was negatively related to the level of symptoms: $B$ = -0.04, $t$ = -2.43, $p < .05$. Table 4 additionally presents mean values of life satisfaction and GHQ-28 scores in two extracted groups of participants.

## Discussion

The results of our study are in accordance with the first hypothesis, as we managed to observe six profiles of participants with regard to scope of knowledge about WWII-related traumatic experiences among ancestors. Particularly, what was characteristic in our sample was the general *lack of knowledge* about this kind of trauma, as the only profile (profile 3) with relatively good memory of such trauma made up only about 14% of the total studied sample. On the one hand, this finding may be significant in itself as this is one of the first studies to address the problem of *heterogeneity* with the aid of latent profile analysis in research on intergenerational trauma [see for review 2, 11]. The dominance of the variable-centred analysis, which neglects how particular subgroups of participants may cluster across the studied predictors and outcomes, may be the possible reason for many conflicting results in the field of intergenerational trauma studies (see introduction). Until now, the person-centred perspective was applied only in one study devoted to the effects of parental Holocaust-related communication and secondary traumatization [5]. Particularly, Shrira [5] using latent profile analysis observed two profiles of offspring of Holocaust survivors with different parental Holocaust-related communication, i.e., offspring who declared intrusive parental communication about the Holocaust perceived themselves as aging less successfully and experienced more anxiety in comparison to offspring who declared informative parental communication. Thus, both our and Shrira [5] studies point to significant obstacles in sharing with subsequent generations the traumatic history of ancestors from WWII. However, whereas our study tackled the problem of vague knowledge about ancestor's trauma, Shrira [5] results pointed to fragmented traumatic knowledge which was transmitted to offspring in intrusive ways. This latter issue, i.e., maintaining a specific conspiracy of silence over trauma experienced in the family, is mentioned in Danieli's [39] *Trauma and the Continuity of Self: A Multidimensional, Multidisciplinary Integrative framework (TCMI)* as one of the major trauma transmission mechanisms through subsequent generations.

Our findings occurred to be also consistent with two other research hypotheses: namely, not only were we able to observe the differences in SWB across profiles of participants, but we

**Table 4. Mean values of GHQ-28 scores and life satisfaction depending on the level of not knowing about the WW-II related traumatic events.**

| | Class 3 | | Class 5 | | | | | |
|---|---|---|---|---|---|---|---|---|
| | *M* | *SD* | *M* | *SD* | *t* | *df* | *p* | *d* |
| **Explained variable** | | | | | | | | |
| Life Satisfaction | 23.10 | 6.54 | 20.54 | 5.76 | 3.01 | 241 | .003 | .43 |
| Somatic Symptoms | 8.39 | 4.77 | 9.17 | 4.14 | -1.27 | 244 | .211 | -.18 |
| Anxiety/Insomnia | 8.58 | 6.24 | 8.84 | 4.26 | -0.31 | 93.72 | .696 | -.05 |
| Social Dysfunction | 7.54 | 4.31 | 8.51 | 3.18 | -1.93 | 243 | .092 | -.27 |
| Depression | 4.14 | 5.68 | 4.73 | 5.00 | -0.79 | 244 | .417 | -.11 |
| GHQ-28 Total Score | 28.65 | 17.51 | 31.26 | 13.07 | -1.12 | 99.11 | .247 | -.18 |

*M*–Mean Value; *SD*–Standard Deviation; *t*–Independent Samples *t* Test; *df*–Degrees of Freedom; *p*–Statistical Significance Acquired From Bootstrapping; *d*–Cohen's *d* Effect Size Measure.

also found worse SWB in the profile with the greatest lack of knowledge about WWII-related traumatic experiences among ancestors (profile 5), which, incidentally, was the most numerous (about 35% of the total sample). Moreover, the only significant differences in life satisfaction among all six profiles were observed among *extreme* profiles, i.e., profile 3, with a level of memory of WWII trauma in the family that was atypically good for the whole sample and the highest life satisfaction, versus profile 5, with the worst life satisfaction of all extracted profiles. In interpreting these results, the aforementioned TCMI framework and the concept of conspiracy of silence may be useful [15, 39]. Specifically, according to this theory, after massive trauma in the family, there is a tendency to preserve silence about it, which stems from two sometimes conflicting tendencies. First, there is the will to protect other family members, especially children, from a knowledge that could be intolerable for them. Second, even if survivors are willing to talk about their traumatic stories, sometimes close relatives indeed cannot bear these often tremendously traumatic stories, so survivors experience denial or lack of social acknowledgement of their trauma. These processes lead to the disruption of the family system and problems in communication and attachment in the family, and thus to further transmission of trauma to subsequent generations, which were all observed in several studies on offspring of Holocaust survivors [6, 17, 30].

Finally, we were able to find one more interesting result among participants belonging to extreme profiles with respect to knowledge about the WWII-related traumatic experiences among ancestors, i.e., profiles 3 and 5: namely, life satisfaction mediated the link between belonging to these profiles and mental, physical, and psychosocial well-being described by the GHQ-28 subscales. In other words, participants from profile 5, with the highest lack of knowledge about family WWII, trauma also had the lowest level of life satisfaction, which was indirectly associated with the poorest GHQ-28 scores, especially with regard to the highest intensity of depressive symptoms, among all profiles of participants. This finding may not only be interpreted in light of the already-mentioned TCMI framework [39], but can also be understood in the context of the perspective of child development, highlighting that sharing family history with children and adolescents helps establish their self-identity and is crucial for their well-being in adulthood [33].

## Strengths and limitations

This theory-driven study was conducted on a large sample of young adults and adopted a person-centred perspective, and these factors represent the strengths of our research. However, several limitations should be also underscored. First, our sample was homogenous with regard to psychosocial functioning, i.e., these were rather highly functioning young adults from large cities in Poland. Future studies could concentrate on more diverse populations in this regard and could also explore the differences between nonclinical and clinical samples (e.g., people suffering from mental disorders). Second, we did not ask participants about the details of family relationships, bonds, and especially communication over the traumatic WWII events of their ancestors. Finally, our study is cross-sectional, which is unfortunately typical for research on intergenerational trauma [2], and prospective studies on this problem remain scarce [40]. In light of this limitation it is also not possible to determine whether participants' life satisfaction actually preceded GHQ-28 scores or whether it was the other way around.

## Conclusion

Despite these limitations, our study adds to the literature on intergenerational trauma by applying a person-centred perspective, which shows that long-term impact of knowledge about traumatic WWII family history may be related to the subjective well-being of young

adults in different ways. In addition, our findings are especially important for Polish literature on trauma, as they highlight the general lack of knowledge of young people about the traumatic WWII experiences of their ancestors. We believe that it can serve as a stimulus for a more comprehensive debate on WWII trauma in Polish society.

## Supporting information

**S1 File. Survey Polish.**
(DOCX)

**S2 File. Survey English.**
(DOCX)

**S3 File.**
(SAV)

## Author Contributions

**Conceptualization:** Marcin Rzeszutek, Maja Lis-Turlejska, Małgorzata Pięta, Monika Karlsen, Holly Backus, Wiktoria Florek, Katarzyna Lisowska, Daniel Pankowski, Szymon Szumiał.

**Data curation:** Marcin Rzeszutek, Małgorzata Pięta, Monika Karlsen, Holly Backus, Wiktoria Florek, Katarzyna Lisowska.

**Formal analysis:** Marcin Rzeszutek, Maja Lis-Turlejska, Szymon Szumiał.

**Investigation:** Marcin Rzeszutek.

**Methodology:** Marcin Rzeszutek, Maja Lis-Turlejska, Daniel Pankowski, Szymon Szumiał.

**Supervision:** Marcin Rzeszutek.

**Writing – original draft:** Marcin Rzeszutek, Maja Lis-Turlejska, Małgorzata Pięta.

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
