## [Decision Letter · Decision Letter 0]

11 Jun 2020

PONE-D-20-05928

Knowledge about Traumatic World War II Experiences among Ancestors and Subjective Well-Being of Young Adults: A Person-Centred Perspective

PLOS ONE

Dear Dr. Rzeszutek,

Thank you for submitting your manuscript to PLOS ONE. After careful consideration, we feel that it has merit but does not fully meet PLOS ONE’s publication criteria as it currently stands. Therefore, we invite you to submit a revised version of the manuscript that addresses the points raised during the review process.

We look forward to receiving your revised manuscript.

Kind regards,

Geilson Lima Santana, M.D., Ph.D.

Academic Editor

PLOS ONE

Journal Requirements:

2. Please include additional information regarding the survey or questionnaire used in the study and ensure that you have provided sufficient details that others could replicate the analyses.

For instance, if you developed a questionnaire as part of this study and it is not under a copyright more restrictive than CC-BY, please include a copy, in both the original language and English, as Supporting Information.

4. Thank you for stating the following in the Title Page of your manuscript:

'Funding: The study was co-financed from the funds of internal funds of the Faculty of Psychology, University of Warsaw and Faculty of Psychology, SWPS University of Social Sciences and Humanities.'

'The funders had no role in study design, data collection and analysis, decision to publish, or preparation of the manuscript.'

5. Your ethics statement must appear in the Methods section of your manuscript. If your ethics statement is written in any section besides the Methods, please move it to the Methods section and delete it from any other section. Please also ensure that your ethics statement is included in your manuscript, as the ethics section of your online submission will not be published alongside your manuscript.

6. Please include captions for your Supporting Information files at the end of your manuscript, and update any in-text citations to match accordingly. Please see our Supporting Information guidelines for more information: http://journals.plos.org/plosone/s/supporting-information

Reviewers' comments:

Reviewer's Responses to Questions

**Comments to the Author**

1. Is the manuscript technically sound, and do the data support the conclusions?

Reviewer #1: Yes

2. Has the statistical analysis been performed appropriately and rigorously? 

Reviewer #1: Yes

3. Have the authors made all data underlying the findings in their manuscript fully available?

Reviewer #1: No

4. Is the manuscript presented in an intelligible fashion and written in standard English?

Reviewer #1: Yes

5. Review Comments to the Author

Reviewer #1: This is an interesting and well-written manuscript. The main findings show that there was variance in the amount of knowledge descendants had about their grandparents WWII-related experiences. Moreover, the less knowledge the descendants had, the lower their subjective well-being was. I have several comments which I hope will help the authors to improve their study. (1) The mediation analyses come as a surprise, because the Introduction and hypotheses say nothing about it. Another problem with this approach is that the current study design is cross-sectional, so it is not possible to determine whether subjective well-being actually precedes GHQ-28 scores or whether it is the other way around. (2) In case the authors decide to maintain the current mediation approach they need to refer to this effect in their literature review and hypotheses and further discuss the limitations of running such an analysis in a cross-sectional design. (3) Please avoid the use of terms such as "full mediation", as such terms may confuse (cf. http://dx.doi.org/10.1016/j.brat.2016.11.001). (4) The authors rightly claim that their study is among the few that took a person-centered approach in the field of intergenerational transmission of trauma. However, they should devote more space to discuss Shrira's (2016) findings, as his study also used latent class analysis trying to delineate communication about and knowledge of ancestral trauma. The authors can compare their own findings to those reported in Shrira (2016). One interesting point arising from such a comparison is that their study mainly tackle the problem of vague knowledge about ancestral trauma, while Shrira (2016) seem to assess fragmented knowledge which was transmitted to offspring in intrusive ways. (5) It is important to clarify which generation the current study relates to (from the age range and the fact that respondents were asked to refer to grandparents' traumatic experiences, I understand the sample consisted of grandchildren of Polish survivors. (6) Please present more details about the group comparison in SWB and GHQ-28. I would like to see the means, SD, and statistical tests including effect size. (7) The Introduction begins by referring to rather early works from the 80s and 90s noting higher level of physical health problems among descendants of Holocaust survivors. This is interesting and I was not aware of these findings (i.e., cancer, heart disease and chronic pain). Can you please provide more details about these findings? (8) Later in the Introduction, when discussing findings about the third generation of Holocaust survivors the authors mention Danieli et al. (2016); however I believe that study investigated second generation only. (9) The rationale mentioned in the first paragraph of the Discussion section should be highlighted in the Introduction section (e.g., using person-centered approach due to conflicting findings in the field of intergenerational transmission).

6. PLOS authors have the option to publish the peer review history of their article (what does this mean?). If published, this will include your full peer review and any attached files.

Reviewer #1: No

---

## [Author Response · Author response to Decision Letter 0]

24 Jun 2020

Dear Editor, Dear Reviewers, 

thank you very much for your suggestions and remarks concerning our article titled “Knowledge about Traumatic World War II Experiences among Ancestors and Subjective Well-being of Young Adults: A Person-Centred Perspective”, which we would like to publish in PLOS One. We referred to all reviewers’ remarks. Below we cite every remark and comment of the reviewers and provide the answers to them in parentheses. All the changes in the revised text are marked with red font.

Editorial comments 

 [Thank you very much for reminder. We double checked that our manuscript meets PLOS ONE's style requirements.]

[Thank you very much. In the Method section we provided all the details regarding the survey we used, which checked their knowledge of traumatic events that participants’ ancestors may have experienced during WWII. We included the copy of the Polish and the English version of this survey in the appendices, in the Supporting Information.]

[It is very important remark. We included our raw data set in the supporting information.] 

[See above. We now included a raw data set in the Supporting Information. We did not know that apart from the possibility of sending data set upon the request, we also have to include it in the Supporting Information.]

[See above.]

4. Thank you for stating the following in the Title Page of your manuscript:

'Funding: The study was co-financed from the funds of internal funds of the Faculty of Psychology, University of Warsaw and Faculty of Psychology, SWPS University of Social Sciences and Humanities.'

[Thank you very much for this remark. We now moved the Acknowledgments section to the Funding Statement.]

'The funders had no role in study design, data collection and analysis, decision to publish, or preparation of the manuscript.' Please include your amended statements within your cover letter; we will change the online submission form on your behalf.

[We removed any funding-related text from the manuscript and we would like to declare that this statement” 'The funders had no role in study design, data collection and analysis, decision to publish, or preparation of the manuscript.' Is OK – we want it to sound exactly like this.]

5. Your ethics statement must appear in the Methods section of your manuscript. If your ethics statement is written in any section besides the Methods, please move it to the Methods section and delete it from any other section. Please also ensure that your ethics statement is included in your manuscript, as the ethics section of your online submission will not be published alongside your manuscript.

[We moved the ethics statement from the title page to the Method section.]

6. Please include captions for your Supporting Information files at the end of your manuscript, and update any in-text citations to match accordingly. Please see our Supporting Information guidelines for more information: http://journals.plos.org/plosone/s/supporting-information

[We included the captions for Supporting Information according to PLOS ONE’s requirements and updated any in-text citations to match accordingly.] 

Reviewers' comments:

Reviewer's Responses to Questions

1. Is the manuscript technically sound, and do the data support the conclusions?

Reviewer #1: Yes

2. Has the statistical analysis been performed appropriately and rigorously?

Reviewer #1: Yes

3. Have the authors made all data underlying the findings in their manuscript fully available?

Reviewer #1: No

 4. Is the manuscript presented in an intelligible fashion and written in standard English?

Reviewer #1: Yes

[Thank you very much for these positive, general, evaluation of our manuscript. As far as the data set is concerned, I would like only to declare that we now included a raw data set in the Supporting Information. We did not know that apart from the possibility of sending data set upon the request, we also have to include it in the Supporting Information.]

Review Comments to the Author

Reviewer #1: This is an interesting and well-written manuscript. The main findings show that there was variance in the amount of knowledge descendants had about their grandparent’s WWII-related experiences. Moreover, the less knowledge the descendants had, the lower their subjective well-being was.

[Thank you very much again for this positive outlook on our study.]

 I have several comments which I hope will help the authors to improve their study. (1) The mediation analyses come as a surprise, because the Introduction and hypotheses say nothing about it. Another problem with this approach is that the current study design is cross-sectional, so it is not possible to determine whether subjective well-being actually precedes GHQ-28 scores or whether it is the other way around.

[Thank you very much for this remark. As far as your first concern, we would like to underline the fact that due to mainly explorative character of this study, we were very caution to put direct hypotheses. Thus, we stated only general assumptions that studied sample will be heterogeneous in terms of knowledge about WWII-related traumatic experiences among ancestors and that belonging to a particular profile will be differently related to subjective well-being among participants (especially that lack of knowledge about WWII-related traumatic experiences among ancestors will be related to worse SWB among our participants). This mediation appeared when we looked for our data set in more details and found this quite unexpected finding. In other words, we did not expect initially this mediation, so we did not mention about it in the introduction nor in the current study section – but sometimes in science in general it happens that you find in your data something you did not expect. Secondly, regarding your remark on cross-sectional character of this study, you are absolutely right - it is not possible to determine whether subjective well-being actually precedes GHQ-28 scores or whether it is the other way around. As you see in the discussion, we were very cautious to draw final conclusions. But we included your remark in the limitations section.]

 (2) In case the authors decide to maintain the current mediation approach they need to refer to this effect in their literature review and hypotheses and further discuss the limitations of running such an analysis in a cross-sectional design. 

[Thank you again very much for the remark concerning our mediational approach. As we stated above, this mediation appeared when we looked for our data set in more details and found this quite unexpected finding. In other words, we did not expect initially this mediation, so we did not mention about it in the introduction nor in the current study section, as our study was mainly explorative (which we mentioned in the current study section), so we also could not comment this mediation in light of other studies in the field, as no such study, i.e. with this specific set of variables, this topic and this methodological design (person-centered) appeared in the literature. Please understand us in this context – as we wrote above, sometimes in science in general it happens that you find in your data something you did not expect. Secondly, in the limitations sections we elaborate more on the limitations of running such an analysis in a cross-sectional design, including your remarks.]

(3) Please avoid the use of terms such as "full mediation", as such terms may confuse (cf. http://dx.doi.org/10.1016/j.brat.2016.11.001).

[We eliminated the phrase “full mediation” throughout the manuscript.]

 (4) The authors rightly claim that their study is among the few that took a person-centered approach in the field of intergenerational transmission of trauma. However, they should devote more space to discuss Shrira's (2016) findings, as his study also used latent class analysis trying to delineate communication about and knowledge of ancestral trauma. The authors can compare their own findings to those reported in Shrira (2016). One interesting point arising from such a comparison is that their study mainly tackles the problem of vague knowledge about ancestral trauma, while Shrira (2016) seem to assess fragmented knowledge which was transmitted to offspring in intrusive ways. 

[It is very important remark. According to your suggestion we discussed Shrira's (2016) findings in a more details, especially in light of our findings on vague knowledge vs. Shrira's (2016) findings on fragmented knowledge which was transmitted to offspring in intrusive ways].

(5) It is important to clarify which generation the current study relates to (from the age range and the fact that respondents were asked to refer to grandparents' traumatic experiences, I understand the sample consisted of grandchildren of Polish survivors. 

[Thank you very much for this remark. The current study sample relates to grand (3-rd generation) or great-grandparent (4-th generation) of Polish survivors – we included this information in the Method section.]

(6) Please present more details about the group comparison in SWB and GHQ-28. I would like to see the means, SD, and statistical tests including effect size. 

[Thank you very much for this remark. We included additional Table to the manuscript.]

(7) The Introduction begins by referring to rather early works from the 80s and 90s noting higher level of physical health problems among descendants of Holocaust survivors. This is interesting and I was not aware of these findings (i.e., cancer, heart disease and chronic pain). Can you please provide more details about these findings? 

[Thank you for your interest. There are many hypotheses on how parental trauma may affect next generation mental and physical health. Due to the cross-sectional nature of this study and related limitations, we did not want to elaborate on this issue in details to avoid some speculative remarks taking into an account our study design and assessed variables. However, here I can only mention for you that growing number of studies have underlined biological and (epi)genetic mechanisms linking parental trauma with changes in offspring’s cortisol metabolism compared to offspring of non-traumatized parents (e.g. Yehuda & Bierer, 2008; Yehuda et al., 2005). Alternatively speaking, parental stress (in case of second generation), in a pre- or post-natal period, affects the stress system of offspring leading to epigenetic and cortisol level changes (Betancourt, 2015). This may be responsible for aforementioned problems with physical health. There is also theory pointing to the fact adverse parental communication patterns and poor attachment styles in the family may be responsible for subsequent problems in mental and physical health due to chronic, but non-expressed distress in offspring Letzter-Pouw et al., 2014.]

(8) Later in the Introduction, when discussing findings about the third generation of Holocaust survivors the authors mention Danieli et al. (2016); however I believe that study investigated second generation only. 

[Yes you are right second generation, but I compare Danieli et al. (2016) study with Sagi-Schwartz et al., 2008 metanalysis on other generation. This comparison was was done to point to the discussion whether trauma transmission imposes negative clinical consequences on subsequent generations or not (e.g., compare Bowers & Yehuda, 2016 and van Ijzendoorn et al., 2003) and which generation (the second or third) is affected most by the traumatic experiences of ancestors (e.g., compare Danieli et al., 2016 and Sagi-Schwartz et al., 2008).]

(9) The rationale mentioned in the first paragraph of the Discussion section should be highlighted in the Introduction section (e.g., using person-centered approach due to conflicting findings in the field of intergenerational transmission).

[We put this rationale also in the Introduction section.]

To sum up, I would like to thank Editor and Reviewers for their time and effort. I found all the comments very useful and I believe that they helped me to improve the manuscript quality. I deeply appreciate a chance you gave me to revise and submit it to be considered for publication in PLOS One.

---

## [Decision Letter · Decision Letter 1]

20 Jul 2020

PONE-D-20-05928R1

Knowledge about Traumatic World War II Experiences among Ancestors and Subjective Well-Being of Young Adults: A Person-Centred Perspective

PLOS ONE

Dear Dr. Rzeszutek,

Thank you for submitting your manuscript to PLOS ONE. After careful consideration, we feel that it has merit but does not fully meet PLOS ONE’s publication criteria as it currently stands. Therefore, we invite you to submit a revised version of the manuscript that addresses the points raised during the review process.

We look forward to receiving your revised manuscript.

Kind regards,

Geilson Lima Santana, M.D., Ph.D.

Academic Editor

PLOS ONE

Reviewers' comments:

Reviewer's Responses to Questions

**Comments to the Author**

1. If the authors have adequately addressed your comments raised in a previous round of review and you feel that this manuscript is now acceptable for publication, you may indicate that here to bypass the “Comments to the Author” section, enter your conflict of interest statement in the “Confidential to Editor” section, and submit your "Accept" recommendation.

Reviewer #1: (No Response)

2. Is the manuscript technically sound, and do the data support the conclusions?

Reviewer #1: Yes

3. Has the statistical analysis been performed appropriately and rigorously? 

Reviewer #1: Yes

4. Have the authors made all data underlying the findings in their manuscript fully available?

Reviewer #1: Yes

5. Is the manuscript presented in an intelligible fashion and written in standard English?

Reviewer #1: Yes

6. Review Comments to the Author

Reviewer #1: The authors have responded to most of my comments yet several issues remained unresolved. (1) Regarding the claim that second and third generations reported higher levels of physical health problems like cancer, heart disease, or chronic pain: In their response the authors mention neuroendocrine and epigenetic works. Still it seems like they stretched the conclusions of these studies in their claim for higher physical morbidity. I also don't think Baranovsky or Sigal & Weinfeld provided empirical evidence for higher physical morbidity in these groups. Therefore I ask the authors to rephrase the sentence or alternatively provide empirical support for their claim. (2) Shrira is misspelled as Shira twice; please correct.

7. PLOS authors have the option to publish the peer review history of their article (what does this mean?). If published, this will include your full peer review and any attached files.

Reviewer #1: No

---

## [Author Response · Author response to Decision Letter 1]

21 Jul 2020

Dear Editor, Dear Reviewers, 

thank you very much for another suggestions and remarks concerning our article titled “Knowledge about Traumatic World War II Experiences among Ancestors and Subjective Well-being of Young Adults: A Person-Centred Perspective”, which we would like to publish in PLOS One. We referred to all reviewers’ remarks. Below we cite every remark and comment of the reviewers and provide the answers to them in parentheses. All the changes in the revised text are marked with red font.

Reviewers' comments:

Reviewer's Responses to Questions

Reviewer #1: The authors have responded to most of my comments yet several issues remained unresolved. 

[Thank you very much for kind words. We tried a lot to incorporate all your remarks; however, we see that not all were incorporated completely properly.]

(1) Regarding the claim that second and third generations reported higher levels of physical health problems like cancer, heart disease, or chronic pain: In their response the authors mention neuroendocrine and epigenetic works. Still it seems like they stretched the conclusions of these studies in their claim for higher physical morbidity. I also don't think Baranovsky or Sigal & Weinfeld provided empirical evidence for higher physical morbidity in these groups. Therefore I ask the authors 

to rephrase the sentence or alternatively provide empirical support for their claim. 

[Thank you very much for this remark. It is very important suggestion. As we wrote in the previous response to your comments, there are many hypotheses on how parental trauma may affect next generation mental and physical health. Due to the cross-sectional nature of this study and related limitations, we did not want to elaborate on this issue in details to avoid some speculative remarks taking into an account our study design and assessed variables. And in the introduction we only mentioned that such theories exist. However, at this time, in the second revision of our study we eliminated from the discussion sections all mentioning about neuroendocrine and epigenetic mechanisms to comment our findings. We also eliminated Baranovsky and Sigal & Weinfeld references when writing about higher physical morbidity in these groups. Once again thank you for this remark.]

(2) Shrira is misspelled as Shira twice; please correct.

[Thank you very much for this remark – we corrected this typo throughout the whole manuscript.]

To sum up, I would like to thank again Editor and Reviewers for their time and effort. I found all the comments very useful and I believe that they helped me to improve the manuscript quality. I deeply appreciate a chance you gave me to revise and submit it to be considered for publication in PLOS One.

---

## [Decision Letter · Decision Letter 2]

30 Jul 2020

PONE-D-20-05928R2

Knowledge about Traumatic World War II Experiences among Ancestors and Subjective Well-Being of Young Adults: A Person-Centred Perspective

PLOS ONE

Dear Dr. Rzeszutek,

Thank you for submitting your manuscript to PLOS ONE. After careful consideration, we feel that it has merit but does not fully meet PLOS ONE’s publication criteria as it currently stands. Therefore, we invite you to submit a revised version of the manuscript that addresses the points raised during the review process.

We look forward to receiving your revised manuscript.

Kind regards,

Geilson Lima Santana, M.D., Ph.D.

Academic Editor

PLOS ONE

Additional Editor Comments (if provided):

Dear author, thank you very much for addressing reviewer's suggestions.

Some final details are needed for accepting your manuscript for publication. It is necessary to adapt it according to PlosOne's guidelines:

https://journals.plos.org/plosone/s/submission-guidelines

1. Include page numbers and line numbers in the manuscript file. Use continuous line numbers (do not restart the numbering on each page).

2. PLOS uses “Vancouver” style, numbered and bracketed.

https://journals.plos.org/plosone/s/file?id=80c1/PLOSOne_formatting_sample_main_body.pdf

3.Please follow the structure and the style of the three levels of heading

4. Pay careful attention to figure captions, legends and file naming for figures.

5. Please pay especial attention to table and table citations: Tables should be included directly after the paragraph in which they are first cited.

Reviewers' comments:

Reviewer's Responses to Questions

**Comments to the Author**

1. If the authors have adequately addressed your comments raised in a previous round of review and you feel that this manuscript is now acceptable for publication, you may indicate that here to bypass the “Comments to the Author” section, enter your conflict of interest statement in the “Confidential to Editor” section, and submit your "Accept" recommendation.

Reviewer #1: All comments have been addressed

2. Is the manuscript technically sound, and do the data support the conclusions?

Reviewer #1: Yes

3. Has the statistical analysis been performed appropriately and rigorously? 

Reviewer #1: Yes

4. Have the authors made all data underlying the findings in their manuscript fully available?

Reviewer #1: No

5. Is the manuscript presented in an intelligible fashion and written in standard English?

Reviewer #1: Yes

6. Review Comments to the Author

Reviewer #1: (No Response)

7. PLOS authors have the option to publish the peer review history of their article (what does this mean?). If published, this will include your full peer review and any attached files.

Reviewer #1: No

---

## [Author Response · Author response to Decision Letter 2]

31 Jul 2020

Dear Editor, Dear Reviewers, 

thank you very much for another suggestions and remarks concerning our article titled “Knowledge about Traumatic World War II Experiences among Ancestors and Subjective Well-being of Young Adults: A Person-Centred Perspective”, which we would like to publish in PLOS One. We referred to all reviewers’ remarks. Below we cite every remark and comment of the reviewers and provide the answers to them in parentheses. 

Editor’s comments

1. Include page numbers and line numbers in the manuscript file. Use continuous line numbers (do not restart the numbering on each page).

[We included page numbers and line numbers in the manuscript file.]

2. PLOS uses “Vancouver” style, numbered and bracketed.

[We changed the style to Vancouver throughout the manuscript.]

To sum up, I would like to thank again Editor and Reviewers for their time and effort. I found all the comments very useful and I believe that they helped me to improve the manuscript quality. I deeply appreciate a chance you gave me to revise and submit it to be considered for publication in PLOS One.

---

## [Editor Report · Decision Letter 3]

5 Aug 2020

Knowledge about Traumatic World War II Experiences among Ancestors and Subjective Well-Being of Young Adults: A Person-Centred Perspective

PONE-D-20-05928R3

Dear Dr. Rzeszutek,

We’re pleased to inform you that your manuscript has been judged scientifically suitable for publication and will be formally accepted for publication once it meets all outstanding technical requirements.

Kind regards,

Geilson Lima Santana, M.D., Ph.D.

Academic Editor

PLOS ONE

Additional Editor Comments (optional):

Thank you for addressing the points I´ve shown you.
---

## [Editor Report · Acceptance letter]

12 Aug 2020

PONE-D-20-05928R3 

Knowledge about Traumatic World War II Experiences among Ancestors and Subjective Well-being of Young Adults: A Person-Centred Perspective 

Dear Dr. Rzeszutek:

I'm pleased to inform you that your manuscript has been deemed suitable for publication in PLOS ONE. Congratulations! Your manuscript is now with our production department. 

Kind regards, 

on behalf of

Dr. Geilson Lima Santana 

Academic Editor

PLOS ONE